# Combining Crop Modeling with Remote Sensing Data Using a Particle Filtering Technique to Produce Real-Time Forecasts of Winter Wheat Yields under Uncertain Boundary Conditions

Hossein Zare [1,*], Tobias K. D. Weber [1], Joachim Ingwersen [1], Wolfgang Nowak [2], Sebastian Gayler [1] and Thilo Streck [1]

1 Institute of Soil Science and Land Evaluation, University of Hohenheim, Emil-Wolff-Straße 27, 70599 Stuttgart, Germany; tobias.weber@uni-hohenheim.de (T.K.D.W.); joachim.ingwersen@uni-hohenheim.de (J.I.); sebastian.gayler@uni-hohenheim.de (S.G.); tstreck@uni-hohenheim.de (T.S.)
2 Institute for Modelling Hydraulic and Environmental Systems, University of Stuttgart, Pfaffenwaldring 5a, 70569 Stuttgart, Germany; wolfgang.nowak@iws.uni-stuttgart.de
* Correspondence: hossein.zare@uni-hohenheim.de

**Abstract:** Within-season crop yield forecasting at national and regional levels is crucial to ensure food security. Yet, forecasting is a challenge because of incomplete knowledge about the heterogeneity of factors determining crop growth, above all management and cultivars. This motivates us to propose a method for early forecasting of winter wheat yields in low-information systems regarding crop management and cultivars, and uncertain weather condition. The study was performed in two contrasting regions in southwest Germany, Kraichgau and Swabian Jura. We used in-season green leaf area index (LAI) as a proxy for end-of-season grain yield. We applied PILOTE, a simple and computationally inexpensive semi-empirical radiative transfer model to produce yield forecasts and assimilated LAI data measured in-situ and sensed by satellites (Landsat and Sentinel-2). To assimilate the LAI data into the PILOTE model, we used the particle filtering method. Both weather and sowing data were treated as random variables, acknowledging principal sources of uncertainties to yield forecasting. As such, we used the stochastic weather generator MarkSim® GCM to produce an ensemble of uncertain meteorological boundary conditions until the end of the season. Sowing dates were assumed normally distributed. To evaluate the performance of the data assimilation scheme, we set up the PILOTE model without data assimilation, treating weather data and sowing dates as random variables (baseline Monte Carlo simulation). Data assimilation increased the accuracy and precision of LAI simulation. Increasing the number of assimilation times decreased the mean absolute error (MAE) of LAI prediction from satellite data by ~1 to 0.2 $m^2/m^2$. Yield prediction was improved by data assimilation as compared to the baseline Monte Carlo simulation in both regions. Yield prediction by assimilating satellite-derived LAI showed similar statistics as assimilating the LAI data measured in-situ. The error in yield prediction by assimilating satellite-derived LAI was 7% in Kraichgau and 4% in Swabian Jura, whereas the yield prediction error by Monte Carlo simulation was 10 percent in both regions. Overall, we conclude that assimilating even noisy LAI data before anthesis substantially improves forecasting of winter wheat grain yield by reducing prediction errors caused by uncertainties in weather data, incomplete knowledge about management, and model calibration uncertainty.

**Keywords:** crop model; data assimilation; particle filtering; PILOTE; prediction uncertainty; yield forecast

## 1. Introduction

Timely and accurate crop yield forecasts are important for a wide spectrum of end-users including policy decision-makers, food security organizations, insurance companies,

and individual farmers [1]. Stakeholders are interested in yield predictions at different scales, and in associated uncertainties. Individual farmers focus on attainable yield in their fields, where information about the management is good but variability and uncertainty of weather and soil conditions may be high. Governments and international food security organizations are interested in yield estimates at county, national or regional levels. Beyond the uncertainty of weather and soil conditions, which generally increases with scale, the poor knowledge of agronomic management as well as of crop cultivars grown challenge yield prediction on larger scales [2,3]. Although heterogeneity is not necessarily a source of uncertainty, lacking information about the heterogeneity is, e.g., in the case of soils and forcing data. All these uncertainties propagate through the forecasting model, which commonly leads to large bias and inaccuracy in yield predictions.

One method to reduce errors in model predictions is to use additional sources of information such as remote sensing data. Proxies for various state variables can be estimated and used to update the running model. This procedure is termed data assimilation (DA; [4]). Different state variables have been assimilated into crop models in literature for example: LAI [3,5,6], soil moisture [7], biomass [8], plant nitrogen content [9], and phenological stages [10]. For more information, see Weiss et al. [11], Jin et al. [4], and Dorigo et al. [12].

Data assimilation methods reduce the discrepancy between observed and modelled variables. There are two major data assimilation strategies: 1. *Recalibration/re-initialization methods*: crop model parameters are recalibrated to minimize the difference between observed and simulated state variables using appropriate objective functions, often in a formal maximum-likelihood framework [2,13]. 2. *Updating methods*: These methods combine data that sequentially become available, e.g., from remote sensing, with ongoing model simulations to update corresponding state variables and parameters in real-time. Thus, the distribution of parameter values can change over time, and statistics of predicted variables can shift. The temporal distribution of the remote sensing data, the handling of the related uncertainties, and other decisions on modelling parameter and model uncertainties play an important role in the predictions.

Bayesian approaches are often used for data assimilation. One is a repeated application of Bayesian updating, whenever new data become available. Whereas many methods and algorithms exist, they have a lot in common. The current knowledge (or uncertainty) about relevant system quantities (state variables, parameters, possible errors) is represented as a probability distribution, called prior distribution in the Bayesian context. Whenever new data come in, these distributions are updated to so-called posterior distributions that represent the new state of knowledge. Different methods differ in how they approximate and update the distributions, in required assumptions, and hence inaccuracy, computational efforts, and application range. The most commonly applied data assimilation algorithms in the field of crop modelling are Ensemble Kalman Filters (EnKF; [2,7,14,15]), Four-Dimensional Variational data assimilation (4DVAR) [16], and Particle Filters (PF; [17–20]).

EnKF has received much attention due to its simple concept and easy implementation. It assumes prediction and observation errors to be normally distributed. Posterior density functions are described solely by means and co-variances. Accordingly, applying EnKFs to (nonlinear) systems with non-Gaussian errors may yield poor results. 4DVAR requires high computational power and is challenging to implement [21,22]. PF, also known as the sequential Monte Carlo filter, estimates the posterior density of the state variables given the observed variables and has been successfully applied to crop modelling [18,20,23]. It relies on sequential Bayesian estimation and importance sampling, it is not restricted to Gaussian errors, and can handle nonlinear changes in the system under consideration [4]. Due to the inherent Monte Carlo sampling, PF methods approximate the whole posterior probability distribution, not only mean and variance (as EnKF). The accuracy of the estimates depends on the resampling method in the filtering [22,24].

The success of DA does not only depend on the assimilation algorithm. Additionally, the quality and quantity of the observation data, information about the system under

consideration, the statistical adequacy of assumed distributions to represent uncertainties, and the structure of the process model play major roles in the performance of the overall procedure. Addressing and understanding the uncertainty in each part is crucial. In the following, we discuss three specific aspects:

1.  Determining state variables of crops from remote sensing signals is not straightforward. This is mainly due to image resolution (spatial, temporal, and radiometric), the background soil effect, signal sensitivity saturation, and atmospheric effects [14,25]. In the early growing season, the soil and soil moisture state blur the signals, while in the mid-late growing season the signals become less sensitive to LAI because of leaf overlapping. The latter is called the saturation phenomenon. Systematic errors in the estimated variables will force the crop model to produce unrealistic predictions [2]. It remains to be clarified, how accurate remote sensing data should be to improve yield prediction. To mitigate deficiencies in the satellite signals, researchers have developed various empirical and mechanistic models. In general, mechanistic approaches provide better results but require more input [11], which is typically unavailable on large scale. Among the empirical remote sensing approaches, the Choudhury model [26] converts satellite signals to LAI using a parameter related to the leaf geometry in the canopy. Therefore, it can be adjusted to erectophile and planophile canopies. Provided that the number of evaluated remote sensing images is high Thorp et al. [5] showed that the Choudhury model performs as good as the mechanistic model PROSAIL. However, fine-resolution, cloud-free data from optical sensors are not always available at regular intervals due to unfavorable atmospheric conditions, especially in areas with high cloud coverage during the growing season.

2.  Another part of uncertainty comes from incompletely resolved system information including explanatory variables (inputs and forcing data). Spatial differences in agronomic practices, mainly sowing date, fertilizer application, and crop genotype, are major contributors to input uncertainty [27]. This information is crucial for running crop models, but it is rarely available on a large scale mainly since data acquisition is too costly. They somehow can be estimated either directly or indirectly by satellite observations. Jégo et al. [28] reduced the yield prediction uncertainty associated with crop management by assimilating remote sensing LAI. Seasonal and yearly variation in weather data that control crop phenology, water availability, and photosynthesis have been recognized as the main sources for inter-annual yield variability [29]. Bias and uncertainty of the weather data may be large, but this is often neglected in the literature [30,31]. The real aim of data assimilation is real-time prediction, requiring generated or forecasted weather data. Most research in this field, however, has used data assimilation for hindcasting with measured or gridded deterministic weather data [7,27].

3.  The third large source of uncertainty lies in the structure of crop models and the parameters relating the assimilated state variable to the target variable, in our case grain yield. Data assimilation may significantly improve the prediction of the assimilated variables but not necessarily the prediction of grain yield. The model structural link between LAI and biomass, for example, is commonly strong, but this is less so for grain yield [32]. Also, the time at which LAI information is available, relative to harvest, has an impact. For example, soil moisture and LAI available during the early growing season may be less important for yield forecast accuracy than data available in the mid or late growing season. A weak correlation between assimilated variables and yield, which may be the result of model structure or parameterization or both, does not lead to improved yield prediction even if the uncertainties previously mentioned are fully accounted for. In other words, the crop variable used for data assimilation should contain information about the uncertainty of the parameters and affect the yield [33]. Model structure is also important since the input uncertainty propagates to yield prediction through crop model equations and parameters. Complex, process-based models can explain many processes in detail and provide the

dynamics of many state variables, provided that the required inputs are available. However, in the absence of large-scale agronomic and soil data, simple models with fewer parameters may be more useful, e.g., the PILOTE model [34,35], which we will employ here, or others such as SAFY [36]. PILOTE simulates LAI as a function of temperature and soil water balance using empirical equations. It then computes biomass from LAI, incoming radiation, and radiation use efficiency.

Since the PILOTE model does not require soil data and full agronomic data (sowing date is the only important data), it can be easily used on large scale when only limited data are available.

In this study, we aim to predict the grain yield of winter wheat using a simple crop model integrated with the PF algorithm, while addressing the sources of error mentioned above. We employed the PILOTE crop model introduced above. PILOTE was calibrated independently in each region using measured data to minimize the error caused by parameter estimation. LAI data from two sources were used to inform the PILOTE simulations: In-situ measurements and remote sensing (Choudhury model). A weather generator was applied to generate stochastic weather data from climate forecast models. Weather and sowing dates were treated as random variables to account for the uncertainty of inputs in the regional model application. As baseline, we used a Monte-Carlo simulation with the calibrated PILOTE model, subject to the same uncertainties, but without DA. Across all simulations, the three sources of error outlined above will be quantified and compared by their impacts on the LAI simulations and grain yield predicted by PILOTE.

## 2. Materials and Methods

This section is structured in five parts. Section 2.1 briefly explains the studied regions and data collection. The stochastic weather data and remote sensing data are explained in Sections 2.1.2 and 2.1.3, respectively. The details of data assimilation and the PF algorithm implementation are outlined in Sections 2.2 and 2.3. Finally, the modelling set up and testing for yield prediction using PF is explained in Sections 2.4 and 2.5. For the sake of brevity, the process model PILOTE and details of parameter estimation are presented in Appendix A.

### 2.1. Study Location and Data

2.1.1. Study Sites

This study was conducted for two regions in southwest Germany, Kraichgau (KR) and Swabian Jura (SJ), using data from 2010 to 2017 (Figure 1). KR is one of the warmest regions in Germany with a mean temperature slightly above 9 °C. Annual precipitation varies from 720 to 830 mm. The elevation of this region is between 100 and 400 m above sea level. With an altitude of 700–1000 m, SJ is substantially colder. The mean temperature is 6–7 °C and annual precipitation 800–1000 mm [37]. The major characteristic soil types of the KR and SJ are Stagnic Luvisol and Leptosol, respectively [38].

Six arable fields (14.9–23.6 ha), managed by local farmers, were monitored during the study period. The fields were named after the Eddy Covariance (EC) stations installed in the center of each of the three fields, EC1 to 3 in KR and EC4 to 6 in SJ. Agronomic management such as the selection of crop rotation, cultivars, fertilization, as well as the timing of sowing and harvesting was selected by the farmers. In total, 19 site-years were dedicated for wheat (Table 1). In EC3-2016, three cultivars were mixed by the farmer. Although the weather and other agronomic managements were similar to other years, the collected data was very different from the other site-years. Therefore, we excluded EC3-2016 from the study.

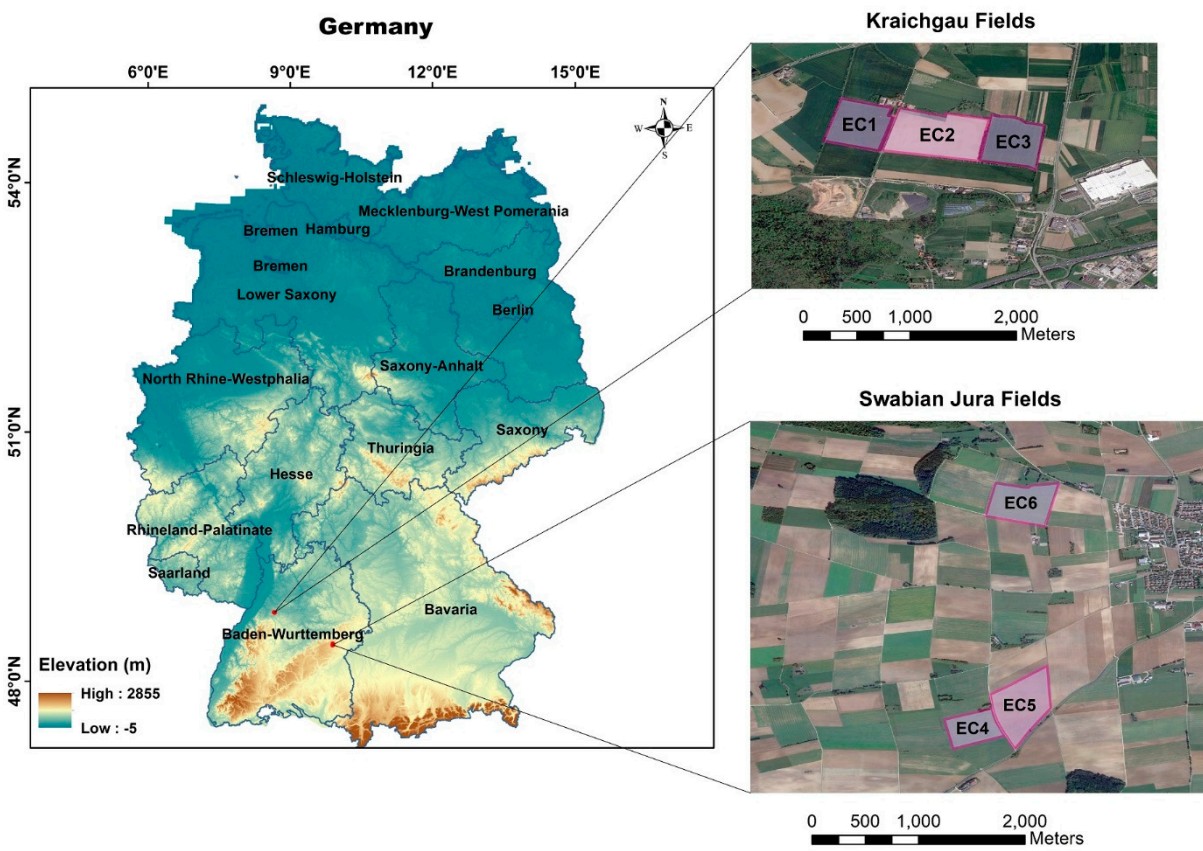

**Figure 1.** Geographical location of research fields in the Kraichgau and Swabian Jura regions, Germany [39].

**Table 1.** Information on winter wheat (*Triticum aestivum*) cultivars (CUL), sowing dates (SD) and harvest dates (HD) for the studied site-years. The empty cells denote periods when other crops were grown.

| Year of Growing Season | EC1 | EC2 | EC3 | EC4 | EC5 | EC6 |
|---|---|---|---|---|---|---|
| **2009–2010** | | | CUL: Cubus SD: 22-Oct HD: 5-Aug | | CUL: Pamier SD: 18-Sep HD: 26-Aug | |
| **2010–2011** | CUL: Akteur SD: 19-Oct HD: 28-Jul | CUL: Akteur SD: 11-Oct HD: 29-Jul | | CUL: Akteur SD: 22-Sep HD: 20-Aug | | CUL: Hermann SD: 13-Oct HD: 22-Aug |
| **2011–2012** | | | CUL: Akteur SD: 18-Oct HD: 1-Aug | | | |
| **2012–2013** | CUL: Akteur SD: 17-Oct HD: 4-Aug | CUL: Akteur SD: 26-Oct HD: 5-Aug | | | | |
| **2013–2014** | | | CUL: JB Asano SD: 25-Oct HD: 4-Aug | CUL: Orcas SD: 8-Oct HD: 23-Aug | | CUL: Pamier SD: 9-Oct HD: 20-Aug |
| **2014–2015** | CUL: Sokal SD: 23-Oct HD: 20-Jul | CUL: Akteur SD: 28-Oct HD: 24-Jul | | CUL: Arezzo SD: 14-Oct HD: 12-Aug | | |
| **2015–2016** | | | CUL: Estivus, Pamier, Ferrum SD: 24-Oct HD: 30-Jul | | | |
| **2016–2017** | CUL: Patras SD: 14-Nov HD: 30-Jul | CUL: Sokal SD: 12-Oct HD: 18-Jul | | | | CUL: Elixer SD: 4-Oct HD: 14-Aug |

Plant measurements were taken bi-weekly. They included plant development stage (BBCH; [40]), plant height, total LAI (non-destructive, using LAI-2000 Plant Canopy Analyzer, LI-COR Biosciences Inc., Lincoln, NE, USA), and aboveground biomass (dried at 60 °C to constant weight). To measure grain yield, generative parts of plants were dried at 28 °C to constant weight. Measurements were taken at five observational plots in each field during the growing period. The mean value of the measurements (mean of the plot means) was used as the representative value for each field. The dataset is publicly available [41]. See this publication also for full details about the experimental procedures. Further details are given by Högy et al. [42], Wizemann et al. [38], Ingwersen et al. [37], Eshonkulov et al. [43], Eshonkulov et al. [44], and Poyda et al. [45]. In this study, we exclusively used *green* LAI, whereas total LAI was measured. Therefore, we used only the LAI measurements before anthesis, assuming that LAI before senescence (LAI until anthesis) equals green LAI. Green LAI measured in-situ is hereafter termed $LAI_{meas}$ (m$^2$/m$^2$).

### 2.1.2. Weather Generator

MarkSim$^{®}$ is a stochastic weather generator widely used in crop modelling. It downscales simulations of General Circulation Models (GCM) and thereby generates stochastic daily weather data [46] from different GCM forecasts. Users can select from many different climate models as well as climate change scenarios. The popular crop modelling system DSSAT routinely uses the MarkSim algorithm to generate stochastic weather data and to fill in missing weather data [46]. The ability to provide fine-resolution weather data (30 arc-second) makes MarkSim useful for small-area applications. Using a third-order Markov process, MarkSim predicts daily minimum and maximum air temperature, daily precipitation, and daily solar radiation values from monthly means of these variables. In this paper, we selected all available climate models under the RCP 4.5 climate scenario to avoid possible biases in the generated weather data. Figure 2 measured and generated daily temperature ensembles (with 99 replications) in KR and SJ in the 2017 growing season.

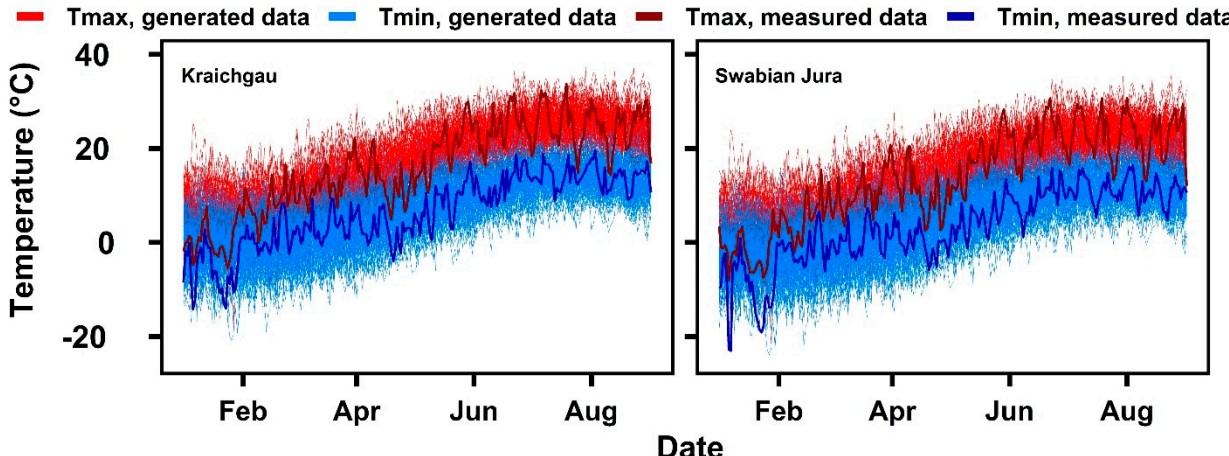

**Figure 2.** Temperature data generated using MarkSim with 99 replicates exemplified for the 2017 wheat growing season. The basis for the generated data is the measurements in 2017 in Kraichgau (**left**) and Swabian Jura (**right**), respectively.

### 2.1.3. Remote Sensing Data

LAI data were derived from satellite data ($LAI_{sat}$). For this, level-1 images obtained from the Landsat 7 and 8 satellites, and from Sentinel-2A and B during the winter wheat growing seasons from 2010 to 2017, were downloaded from the USGS [47] and Copernicus Open Access Hub [48] websites. Level-1 products are radiometrically calibrated with systematic geometric corrections applied. We applied atmospheric correction using the Semi-Automatic Classification plugin included in the software QGIS (v.3.12) [49].

Cloud-free images from March, when wheat LAI starts expanding in spring, until June used in each year and region for data assimilation (2014, 2015, and 2017) are presented in Table 2.

**Table 2.** Information on the satellite images used for data assimilation in the study. Cloud-free images from mid-March to the end of June (wheat anthesis) were selected.

| Year | Kraichgau | | Swabian Jura | |
|------|-----------|----------|--------------|----------|
| | Date | Satellite | Date | Satellite |
| **2014** | 20-Mar | Landsat 8 | 29-Mar | Landsat 7 |
| | 28-Mar | Landsat 7 | 9-Jun | Landsat 7 |
| | 8-Jun | Landsat 8 | | |
| **2015** | 4-Apr | Landsat 8 | 4-Jun | Landsat 8 |
| | 24-Apr | Landsat 8 | 12-Jun | Landsat 7 |
| | 18-May | Landsat 7 | | |
| | 3-Jun | Landsat 7 | | |
| **2017** | 21-Apr | Landsat 7 | 10-Apr | Sentinel-2 |
| | 10-May | Sentinel-2 | 30-Apr | Landsat 7 |
| | 17-May | Sentinel-2 | 10-May | Sentinel-2 |
| | 23-May | Landsat 7 | 17-May | Sentinel-2 |
| | 16-Jun | Landsat 8 | 19-Jun | Sentinel-2 |

The dimensionless Normalized Difference Vegetation Index (NDVI) was calculated using the red and near-infrared bands of the satellite data [50]. Arithmetic averages over the area of each field were used to get a single value for each field from the corresponding image. To avoid mixed pixels on the field borders, we excluded the pixels located on the border of the fields. From NDVI, LAI was calculated following Choudhury et al. [26]. In a first step, the green vegetation fraction $GVF$ (−) was calculated from NDVI:

$$GVF = \frac{NDVI - NDVI_{min}}{NDVI_{max} - NDVI_{min}}, \tag{1}$$

where $NDVI_{max}$ and $NDVI_{min}$ are the maximum and minimum NDVI during the wheat growing period in each pixel of wheat fields. In a second step $GVF$ was converted to LAI by:

$$LAI_{sat} = \frac{\ln(1 - GVF)}{-\beta}, \tag{2}$$

where $\beta$ (-) is a parameter describing the leaf angle distribution that varies between 0.4 to 0.9 in wheat [5]. Then, we used the mean $LAI_{sat}$ for each field to be comparable with the $LAI_{meas}$ data. We estimated parameters $NDVI_{max}$, $NDVI_{min}$ and $\beta$ by minimizing the sum of squared differences between $LAI_{sat}$ and $LAI_{meas}$ (See Appendix A).

### 2.2. Data Assimilation Methodology

Dynamic systems describe how a vector of state variables evolves over time under the influence of explanatory variables (system inputs) and corresponding parameters. In a state-space time-discrete formulation, the generic form of a dynamic model is written as:

$$x_j = f\left(x_{j-1}, \Theta, u_{j-1}\right) + \omega_{j-1}, \tag{3}$$

where $x_j$ is the vector of state variables at time $j$, $f(\cdot)$ signifies the nonlinear dynamic prediction model, $\Theta$ represents a parameter vector, and $u$ are explanatory variables. The prediction model represents the state variable transition from time $j - 1$ to $j$ in response to model inputs (explanatory variables), the parameter vector, and the state variable at time $j - 1$ ($x_{j-1}$). There is usually a discrepancy between modelled and true values of variables even with a perfect dynamic model. The corresponding error in the model function is

represented by $\omega_{j-1}$, which is the stochastic part of the equation. In this system, the current state depends only on the system state of the previous time step. In statistics, this is referred to as a first-order Markov process. Longer memory can be implemented by augmenting the state vector of additional elements that store past information. Note that the parameter vector $\Theta$ is not time-dependent. In our case, the prediction model is the crop (PILOTE) model. The explanatory variables $u$ in crop modelling include forcing data (weather data time series), and agronomic management [32]. Parameters $\Theta$ and other static values and initial conditions are specified via $x_j$ for $j = 1$. All of the initial conditions, explanatory data, parameters and errors can be viewed as random variables (e.g., via stochastic weather simulations). In that case, $x_j$ becomes a random time series of uncertain predictions.

Knowledge about a system's state variable can be estimated by proxies, that is, a different source of information, e.g., from field measurements or satellite data in regular or irregular intervals ($y_j$). The link between state $x_j$ and data $y_j$ is characterized by the noisy observation model:

$$y_j = h(x_j) + v_j, \tag{4}$$

where $h(\cdot)$ is the observation model, and $v_j$ is the error between the observation model and the state variable of interest. In this context, $h(\cdot)$ can either be a remote sensing-based model for estimating LAI such as the Choudhury model (Equation (2)), or a simple field measurement with error $v_j$. Just like $\omega_{j-1}$ in Equation (3), $v_j$ is treated as a random variable.

Both prediction and observation models are necessary to infer the system state by data assimilation. The idea of data assimilation is to apply the observation model to modify the posterior probability prediction of the prediction model [51] using Bayesian inference.

### 2.2.1. The Recursive Bayesian State Estimator

The goal of a Bayesian estimator is to recursively approximate the posterior probability density function (pdf) of a variable at time $j$ and of possible predictions at time $j^+ > j$ conditional on all available observations until time $j$, that is, $p(x_j|Y_j)$, where $Y_j$ is the short-hand notation for $y_1, y_2 \ldots, y_j$.

Here, $p(x_j|Y_j)$ is the posterior pdf of the state variable of interest. Performing the prediction (Equation (3)) and observation models (Equation (4)) in the Bayesian theorem, we can write the conditional pdf of one state variable as:

$$p(x_j|Y_j) = \frac{p(x_j|Y_{j-1}) \; \overbrace{p(y_j|x_j)}^{\text{Observation model}}}{p(y_j|Y_{j-1})}, \tag{5}$$

where $p(y_j|x_j)$ is the likelihood function computed by the observation model. $p(x_j|Y_{j-1})$ is the prior pdf of $x_j$ given all observations until time $j-1$, and $p(y_j|Y_{j-1})$ is the normalizing factor. The prior can be rewritten in integral form:

$$p(x_j|Y_{j-1}) = \int \overbrace{p(x_j|x_{j-1})}^{\text{Prediction model}} p(x_{j-1}|Y_{j-1}) \, dx_{j-1} \tag{6}$$

All pdfs on the right-hand side of Equation (6) can be computed; the transition pdf $p(x_j|x_{j-1})$ is the pdf of the state variable at time $j$ given the states of the previous time step $j-1$. It is computed with the prediction model. The pdf $p(x_{j-1}|Y_{j-1})$ is the posterior of the previous time step. If the initial condition of the state variable $p(x_0)$ is known (at the beginning of simulation when there is no observation), then $p(x_{j-1}|Y_{j-1})$ can be simply computed recursively [51]. Additionally, one can make predictions into the future at time $j^+ > j$ by recursive applications of Equation (6) without any additional data: $p\left(x_j^+|Y_{j-1}\right) = \int p\left(x_j^+|x_j\right) p(x_j|Y_{j-1}) \, dx_j$. In such future steps, all explanatory variables in the prediction models will be replaced by estimates or stochastic predictions.

The normalizing factor $p(y_j|Y_{j-1})$ is the marginal probability of the data that resolves the requirement for a probability distribution to integrate to 1. This can be obtained by evaluating the integral:

$$p(y_j|Y_{j-1}) = \int p(y_j|x_j)p(x_j|Y_{j-1})dx_j, \tag{7}$$

which is exactly the integral over the nominator in Equation (5) to ensure that the resulting $p(x_j|Y_j)$ integrates to unity so that it is a proper pdf. Equation (7) resembles the average (over the previous distribution) goodness of fit (as expressed by the likelihood) of the predictions in the current data assimilation step. Hence, it is often called average likelihood and expresses the goodness of the current pair $p(x_j|Y_{j-1})$ against the incoming data.

All terms of Equation (5) are now available. Except in very few cases under strong assumptions, the Bayesian estimator must be evaluated numerically [51]. To this end, we apply the Particle Filtering (PF) method.

### 2.2.2. Particle Filtering Method

Particle Filtering is a stochastic nonlinear sequential Bayesian filter based on Monte Carlo simulation. At the beginning of the simulation, we generate N realizations, which will propagate through the system as calculated with the prediction model, here PILOTE. These realizations, also called particles, are vectors of states that evolve in time. In fact, each particle contains a vector of states and all the required inputs (explanatory variables) $u_{j-1}$, parameters $\Theta$, and the corresponding noise $\omega_{j-1}$ (Equations (3) and (8)). Different particles can be distinguished by the noise, the parameter vector, or the explanatory variables. Then, particles are propagated through the system using the prediction model. The *ith* particle $x_j^i$ is obtained at time $j$ by:

$$x_j^i = f\left(x_{j-1}^i, \Theta^i, u_{j-1}\right) + \omega_{j-1}^i \quad i = 1, \ldots, N \tag{8}$$

At the beginning of the simulation ($j = 0$), all $x_j^i$ equal zero. Once the first observation is available, the likelihood for each particle ($p(y_j|x_j^i)$) is calculated using the observation model and the state variable of the particle. The likelihood model is selected from the error terms in the observation model ($v_j$ in Equation (4)). The error terms in the observation model (explained in Section 2.3.2) are assumed to follow a normal distribution.

Then, we normalize the likelihoods of all $i = 1, \ldots, N$ particles, $p(y_j|x_j^i)$ to unity and name them $q_j^i$ by:

$$q_j^i = \frac{p(y_j|x_j^i)}{\sum_{i=1}^{N} p(y_j|x_j^i)} \tag{9}$$

This normalization step represents the normalizing effect of Equation (7). These normalized likelihoods $q_j^i$ are used to filter the particles such that particles with higher likelihoods have a larger probability to remain, by. Sequential Importance Resampling (SIR; [51,52]). The particles remaining after resampling are called *surviving particles*. The others, which were not selected in the resampling procedure, are called *killed particles*. The idea of SIR is always to replenish the ensemble size of survived particles back to N, achieved by sampling with replacement. There are several statistical methods available for implementing these random decisions for SIR. In the one we choose, particles with low values of $q_j^i$ will most probably be killed. Particle with large values will probably survive and particles with very high values will probably survive several times. Importantly, survivors can exist several times as identical copies but will diversify again over time due to the future noise ($\omega$) in Equation (3). If particles differ also in parameters, the SIR also affects the parameter distribution effectively updating the model calibration. After the SIR step, the particles represent $p(x_j|Y_j)$, and the entire scheme is repeated in the next assimilation step.

*2.3. Particle Filtering Implementation*

2.3.1. Prediction Model and Driving Uncertainties

In this study, the PILOTE model [34,35] is used as the prediction model. PILOTE is a crop model with a soil and a plant module (Appendix A). In principle, the soil module calculates the water balance on daily time steps using a capacity approach [53]. The core of the plant module is LAI, which in turn determines aboveground biomass and yield. In order to use the PILOTE model as the prediction model (Equation (3)) for the data assimilation, we need to modify (Equation (A2) explained in Appendix A) to the form of Equation (3), that is, the LAI state of PILOTE at time $j$ has to depend on the state at $j-1$. Applying Euler's method this leads to:

$$LAI_{j+1} = LAI_j + \left. \frac{dLAI}{dTT} \right|_j * \Delta TT_j \tag{10}$$

where the derivative $\left. \frac{dLAI}{dTT} \right|_j$ is taken on day $j$ (Equation (A2)). Note that $\Delta TT_j = T_{mean} - T_{base}$ on day $j$. Now, the prediction model may be expressed in the form of Equation (3):

$$LAI_j = f\left(LAI_{j-1}, \Theta, u_{j-1}\right) + \omega_{j-1} \quad with \ j > 1, \tag{11}$$

where the statistics of $\omega$ were taken from the residuals of the prediction in model calibration.

To perturb particles in their initialization phase ($j = 0$), we used, in addition to the stochastic component $\omega$, common major sources of uncertainty for yield predictions on the large scale. These include: sowing date, weather data ($u$), and parameters ($\Theta$). Typical sowing dates in each region (from the dataset) were used to define the bounds of the sowing date time periods. A normal distribution with the mean of 25 October for KR and 7 October for SJ and a standard deviation of one week were used. Weather data were stochastically generated using MarkSim (see Section 2.1.2). The two parameters $TT_f$ and $LAI_{max}$ in the PILOTE model (Equation (A2)) were considered uncertain because they are genetic parameters, meaning that their values are cultivar dependent. The corresponding values were sampled from posterior distributions from the calibration. Finally, the particles were generated as described above (Section 2.2.2).

2.3.2. Observation Model and Likelihood

As we are using two types of data, we need to build two different observation models and two different likelihood functions. The first data type is LAI field data from the five observational plots. In this case, the observation model is direct measurement corrupted by an additive error with normal distribution, zero mean, and an adjustable variance. We used the empirical variance from repeated measurements to select the variance for $v_j$ (Equation (4)). The second data type is the satellite-derived LAI data. As described above, we use the Choudhury model (Section 2.1.3) as the observation model. In order to capture both the noise in the data and possible approximation errors of the Choudhury model, we calibrated the Choudhury model by fitting its satellite-derived LAI values to our own field LAI data. We then estimated the statistics of the residuals. The residuals indicated a normal distribution, again leading us to select a normal likelihood function. To select its variance $v_j$, we used the residual variance after calibration. In order to include the additional uncertainty of noise in the field-LAI data used for calibration, we increased this value for $v_j$ by the noise variance of the field-LAI data discussed above.

Thus, overall, both data types follow a Gaussian (normal) likelihood model:

$$L = \frac{1}{\sigma\sqrt{2\pi}} \exp\left( -\frac{(LAI_{obs} - LAI_{sim})^2}{2\sigma^2} \right), \tag{12}$$

in which $L$ is the Gaussian likelihood of the observed LAI ($LAI_{obs}$) given the simulated LAI ($LAI_{sim}$). Here, $\sigma$ represents the standard deviation of the observations that is taken from the $v_j$ in the respective observation model (field or satellite).

### 2.4. Data Assimilation Setup

We implemented two different data assimilation approaches, one based on the LAI measured in-situ (DA$_{meas}$) and the other one using LAI from satellite remote sensing (DA$_{sat}$). Moreover, the effect of the number of observations on PF performance was examined by applying data assimilation in different steps. The total number of steps was limited by the number of satellite data available during the wheat growing period that is provided in Table 2. In the first step, only the first measurement/satellite dataset was assimilated, and the entire remaining time was simulated up to harvest without further assimilation. Then, in the second step, the first and second datasets were assimilated. In the third step, three data were assimilated and so forth until the last available data. An example of DA$_{sat}$ steps is given in Appendix B.

To account for the uncertainty of future weather in this analysis, measured weather data (past and present) and stochastic weather data were combined to run the models. In assimilation step one, measured weather data were applied to run models until the first observation; then for the rest of the growing season, the stochastic weather generator was used. In the second step, measured weather data were used to run the models until the second observation; afterwards, the stochastic weather generator was used to complete the growing season and so forth. Whenever some current part of weather is switched to measured data the stochastic weather generator is re-run to reflect the newly available (current) weather data as initial conditions.

### 2.5. Testing against Monte-Carlo Simulations

The DA setup was applied using data from nine site years from 2014 to 2017. The PILOTE model simulations (without data assimilation) were used with the same statistical initialization conditions as in the DA setup. We call it Monte Carlo (MC) simulations. Thus, sowing dates, weather data and posterior distributions of the parameters $TT_f$ and $LAI_{max}$ were also taken uncertain in the MC simulations. The DA results were tested against the MC results. The flowchart of the data assimilation trajectories and Monte Carlo simulation is shown in Figure 3. Hereafter, DA approaches using $LAI_{meas}$ and $LAI_{sat}$ and Monte Carlo simulation are called DA$_{meas}$, DA$_{sat}$ and MC, respectively. To statically compare the results, we calculated Root Mean Square Errors (RMSE), Bias and Mean Absolute Errors (MAE) between the simulations and the mean measurements. We should note that the $LAI_{sat}$ was not used for RMSE and Bias calculations. All the analyses including running the PILOTE model, calibration, and data assimilation implementation were performed in the R programming environment. We used the following R packages: Evapotranspiration [54], FME [55], hydroGOF [56], raster [57], and sf [58].

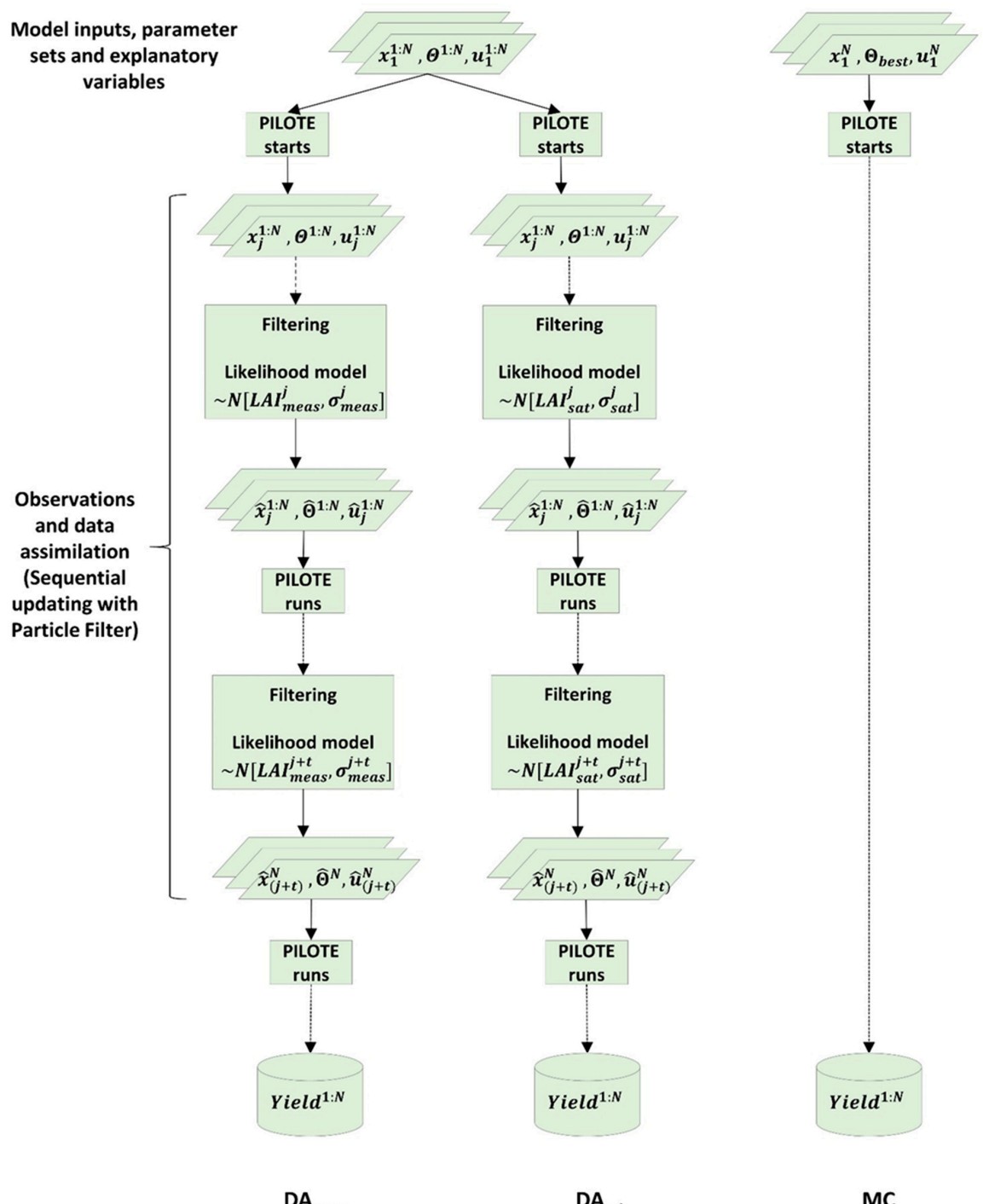

**Figure 3.** Yield prediction process with different methods. The two approaches on the left differ in the filtering technique applied. $LAI_{meas}$ and $LAI_{sat}$ are measured and remotely-sensed LAI, respectively with $\sigma_{obs}$ and $\sigma_{sat}$ represent the corresponding standard deviations. The symbols $x$, $\Theta$, $u$ and $N$ represent state variables, parameter vector, explanatory variables (initial conditions and forcing data), and the number of particles. $\hat{x}$, $\hat{\Theta}$ and $\hat{u}$ are the corresponding filter-updated counterparts. $\sim N\left[LAI_{meas}^{j}, \sigma_{obs}^{j}\right]$ stands for the normal distribution with mean $LAI_{meas/sat}^{j}$ and standard deviation $\sigma_{meas/sat}^{j}$. The sequential updating continues until the last observation is available. $DA_{meas}$ stands for data assimilation based on LAI data measured in-situ, $DA_{sat}$ for data assimilation based on LAI from satellite observations and MC for Monte-Carlo simlations as baseline to test the performance of the data assimilation methods.

## 3. Results

PILOTE was calibrated successfully (Appendix A). Hindcasting with the calibrated model using measured data without uncertainty was similar to the calibration, which shows that parameter estimation was reliable. In KR, RMSEs of yield simulation for the calibration and evaluation were lower than 600 and 910 kg/ha, respectively. These values correspond to relative errors of ~6.5% and ~9.6%, respectively. The bias of yield simulation in KR was lower than 300 kg/ha for calibration and evaluation. In SJ, RMSEs of yield simulation were 323 kg/ha (3.6% error) and 780 kg/ha (8% error) at calibration and evaluation, respectively. Biases at calibration and evaluation were 209 and 291 kg/ha. It is worth noting that only a good calibration enables an unbiased comparison between the DA and MC simulation because only then the improvement in the DA is due to a reduction in model parameters and input uncertainties.

### 3.1. Real-Time LAI Prediction

The Mean Absolute Errors (MAE) of the predicted LAI under uncertain model inputs (sowing date and weather data) are presented in Figure 4. The points show the mean MAE as an indicator for the prediction accuracy and the corresponding bars represent the standard deviation of the MAE which is representative for the uncertainty (precision) of LAI prediction. $DA_{meas}$ predictions were the more accurate (smaller bias) and precise (smaller variance). The lowest and highest MAE acquired by $DA_{meas}$ were 0.15 m$^2$/m$^2$ and 1.10 m$^2$/m$^2$ in EC6_2017 and EC4_2014, respectively. The mean and standard deviation of the MAE decreased in most site-years as the number of assimilation steps increased (except EC2_2015). LAI prediction by $DA_{sat}$ was close to that of $DA_{meas}$ in most cases, but due to the irregular availability of satellite images, the step-by-step comparison is difficult. The largest difference in LAI prediction between $DA_{meas}$ and $DA_{sat}$ can be seen in EC6_2014 where MAE in predictions by $DA_{sat}$ was 0.70 m$^2$/m$^2$ and it was less than 0.20 m$^2$/m$^2$ by $DA_{meas}$ after two assimilation steps.

In contrast, LAI predictions by MC showed a lower predictive power than those obtained by DA. The mean MAE of the predictions produced by MC varied between 0.50 and 1.40 m$^2$/m$^2$. This is not far from the error of LAI in the PILOTE validation result, which means that the stochastic sowing dates and weather data do not substantially affect LAI prediction accuracy. The lowest and highest MAEs were found in EC2_2017 and EC6_2017 (Figure 4).

The standard deviation of MAE represents the uncertainty of the predictions. As can be seen in Figure 4, uncertainties often decline along with the assimilation step in both DA approaches ($DA_{meas}$ and $DA_{sat}$), whereas MC simulations provided a highly uncertain prediction. Recall that uncertainty of LAI predicted by DA originates from sowing date and weather data as well as the model parameters ($TT_f$ and $LAI_{max}$), in addition to the model noise. With increasing assimilation steps, the prediction uncertainty declines because inconsistent parameter combinations are eliminated. Assimilating $LAI_{meas}$ reduced the uncertainty more than assimilating satellite observations. This is a direct consequence of the corresponding error variance in the likelihood function (Equation (12)). The parameter $\sigma$ in the likelihood function reflects the uncertainty of observations (see Section 2.3.2) which is commonly larger for satellites than for in-situ measurements. Therefore, unfit parameter realizations are punished less harshly.

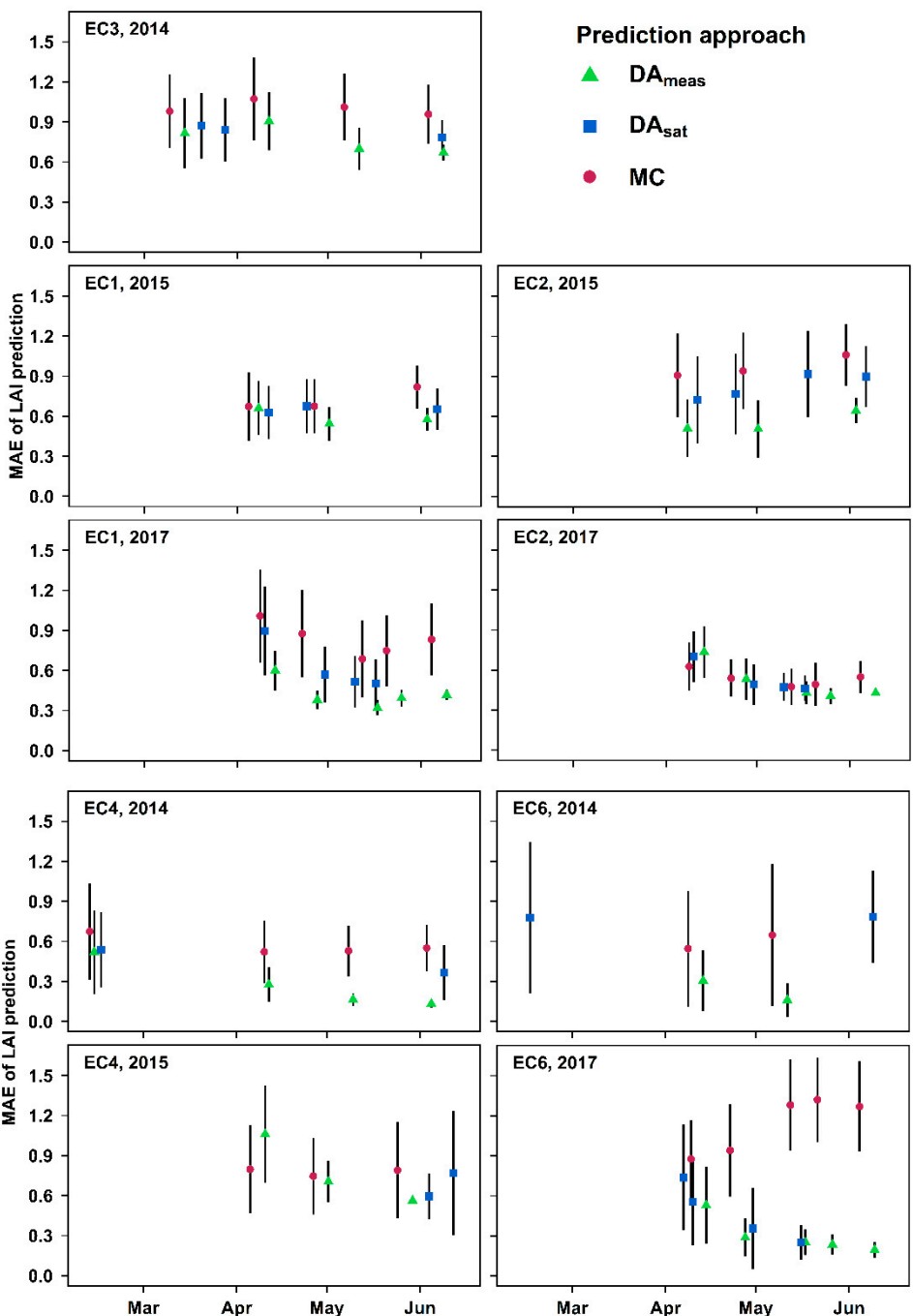

**Figure 4.** Mean Absolute Error (MAE) of LAI predicted with data assimilation and Monte Carlo simulation. Points and bars represent mean and standard deviation MAE of prediction ensembles. Red, green, and blue points show MC (Monte Carlo simulation), DA$_{meas}$ (data assimilation based on data measured in-situ), and DA$_{sat}$ (data assimilation based on satellite observations).

### 3.2. Real-time Grain Yield Prediction

Figures 5 and 6 show the prediction of grain yields. Measured grain yield ranged roughly from 6000 kg/ha to 12,000 kg/ha across all sites and years. In general, as the number of assimilation steps increased, the predictions became more accurate (the median of predictions approached the median of measurements). In most cases, the yield increased with the number of assimilation steps, due to LAI assimilation (right column and middle column in Figures 5 and 6) and the increase in the proportion of measured weather data compared to stochastic weather data in the simulations (MC simulations). We consider

the prediction accuracy satisfying if the median of the predictions (blue boxes) matches the interquartile range (IQR, middle fifty percent) of the measurements (red boxes) (see Figures 5 and 6). In this sense, DA$_{meas}$ almost achieved this goal in KR except for EC2_2015 where the predictions were slightly underestimated (the distance between the median of the prediction and the IQR lower band of the measurements was about 5 percent). In SJ, DA$_{meas}$ predictions underestimated yield in EC4_2014 and EC6_2014. We note that in EC4_2014 measurements varied very little. In the case of EC6_2014, the low prediction quality could be explained by the availability of only two available LAI measurements in April and May for DA$_{meas}$. Due to the variation in LAI, only two measurements are not enough to robustly calculate an average. We also note that increasing the number of steps led to a decrease in the spread of whiskers of the prediction boxes, which indicates higher precision in the predictions (Figures 5 and 6).

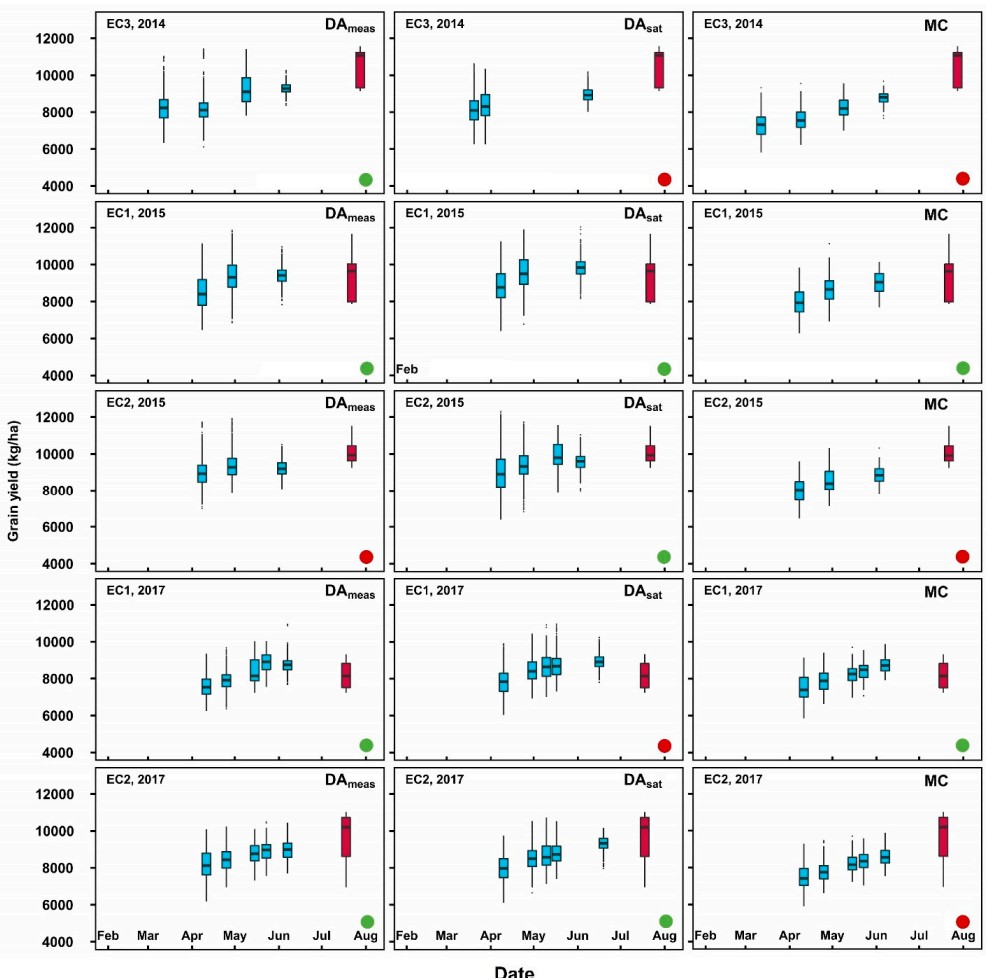

**Figure 5.** Wheat grain yield prediction with data from Kraichgau (KR). **Left column**: DA$_{meas}$ (data assimilation based on LAI measured in-situ), **middle column**: DA$_{sat}$ (data assimilation based on satellite observations of LAI) and **right column**: MC (Monte Carlo simulation). The red boxes show the measured yield. The dates of the blue boxes in DA$_{meas}$ and MC subplots show the dates of in-site measured LAI, and the dates of the blue boxes in DA$_{sat}$ subplots show the dates of remote sensing LAI. The dates of the red boxes correspond to the harvest date. Subplots with green circles signify that the median prediction of the last assimilation step is within the interquartile range of the measurements, otherwise, subplots have red circles.

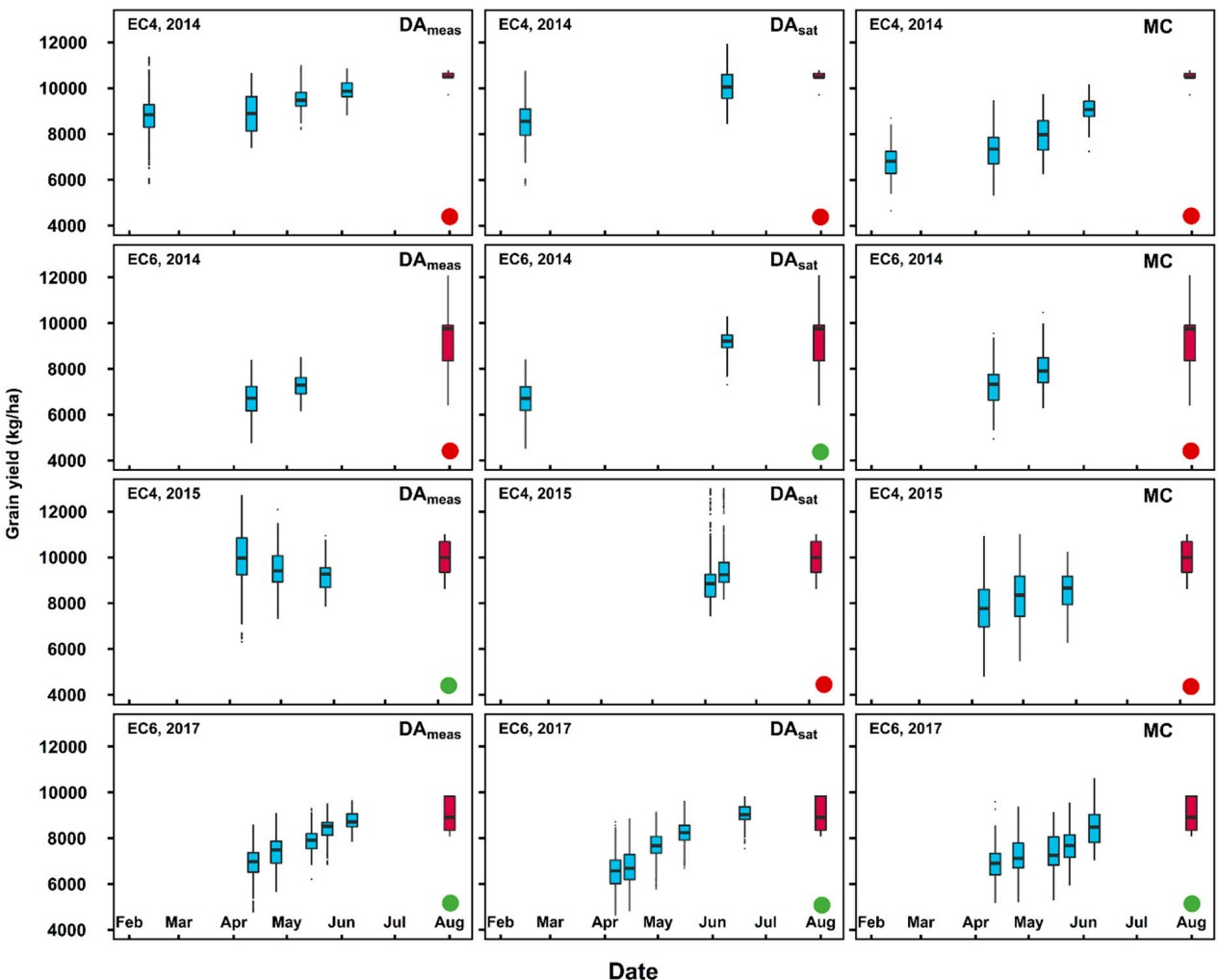

**Figure 6.** Wheat grain yield prediction with data from Swabian Jura (SJ). **Left column**: DA$_{meas}$ (data assimilation based on LAI measured in-situ), **middle column**: DA$_{sat}$ (data assimilation based on satellite observations of LAI) and **right column**: MC (Monte Carlo simulation). The red boxes show the measured yield. The dates of the blue boxes in DA$_{meas}$ and MC subplots show the dates of in-site measured LAI, and the dates of the blue boxes in DA$_{sat}$ subplots show the dates of remote sensing LAI. The dates of the red boxes correspond to the harvest date. Subplots with green circles signify that the median prediction of the last assimilation step is within the interquartile range of the measurements, otherwise, sublopts have red circles.

DA$_{sat}$ performed similarly to DA$_{meas}$, both in the prediction quality and in so far as that predictions became more accurate as the number of assimilation steps increased. The major difference between DA$_{sat}$ and DA$_{meas}$ can be found in EC3_2014, where the median of the predictions was 1200 kg/ha lower than the median of measurements, and in EC1_2017, where it was 800 kg/ha lower. The predictive power of the DA approaches is higher than that of the baseline MC simulations. With MC, we can see underestimations in EC3_2104, EC2_2015, EC4_2014, EC6_2104, and EC4_2105. Here, the median of MC predictions did not fall into the IQR of the measurements. The second drawback of MC was in the prediction uncertainty (reflecting the effect of sowing dates and stochastic weather data on yield), particularly in SJ. There, the variation of predictions is much higher than that of the measurements (Figure 6).

To give a summarizing picture of the performance of DA prediction compared to the baseline MC simulations, we aggregated all site-years of each region and plotted the error in grain yield prediction (difference between predicted and measured yield) for the main

plant growth months leading to anthesis (Figure 7). Not unexpectedly, the predictions in April were worse than in June because LAI is very low at that time and has a low effect on the final yield. Prediction errors by DA were lower in May and June compared to MC. Prediction error in June dropped substantially in SJ while it remained almost flat in KR. DA$_{meas}$ and DA$_{sat}$ performed similarly in both regions. In summary, DA prediction error in June was about 8 and 4 percent in KR and SJ, whereas MC prediction error was about 10 percent in both regions.

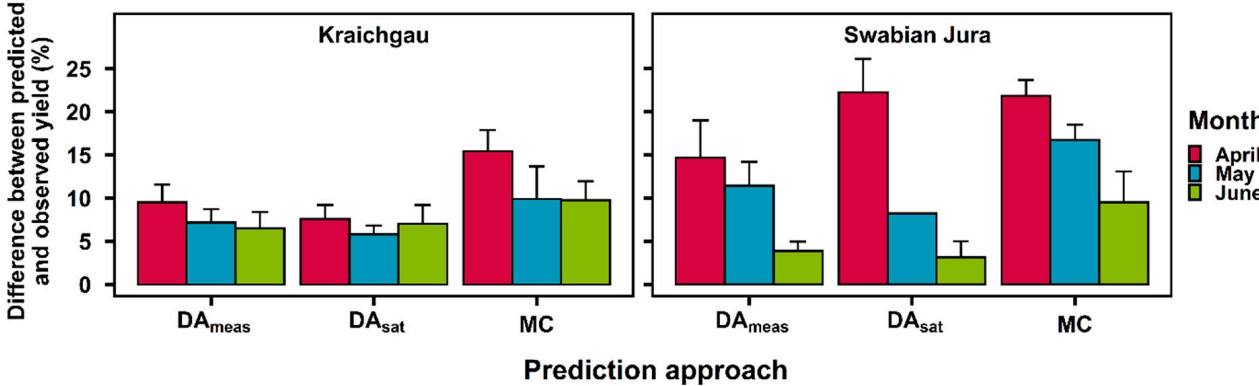

**Figure 7.** Grain yield prediction error (%) from April to June in Kraichgau and Swabian Jura. DA$_{meas}$, DA$_{sat}$ and MC stand for data assimilation based on LAI measured in-situ, data assimilation based on satellite observations of LAI, and Monte Carlo simulation, respectively. Bars show the standard deviation. Grain was harvested in the end of July/first days of of August (Kraichgau) and in August (Swabian Jura) (Table 1).

## 4. Discussion

The success of DA generally depends on the assimilation protocol, uncertainty considerations [59], observation errors, time of assimilation [2], and the relationship between assimilated variables and the target variable (here grain yield) [32]. We mainly focused on the uncertainty (or error) in the assimilation protocol (prediction and observation models), observation data, and inputs including sowing dates and weather data.

### 4.1. Error in the Assimilation Protocol and Observation Data

With the proper assimilation protocol and observation data, we expected that (1) the model predictions of the assimilated variable would improve in comparison to simple MC simulations; (2) there would be similar results between DA$_{meas}$ and DA$_{sat}$; (3) the predictions of LAI would improve the more data were assimilated.

MAEs of predicted LAI in both DA procedures were often lower than those of the baseline MC simulations. That is, the DA schemes were successful in generating and retaining particles with higher probabilities to follow measurements. This can be qualified by evaluating the denominator in Equation (9), which is proportional to the average likelihood in Equation (7), and inversely proportional to the number of killed particles.

Although the dates and numbers of available satellite images were not the same as the measurements, the mean MAEs in LAI predictions by DA$_{sat}$ were close to those by DA$_{meas}$. This implies that the assimilation protocol was able to mitigate the errors in $LAI_{sat}$ and prevent propagating them into the predictions. The major difference between LAI predictions by DA$_{meas}$ and DA$_{sat}$ can be seen in the uncertainty of the predicted LAI (MAE bars in Figure 4) which is a direct consequence of the uncertainty considered in the DA$_{sat}$ protocol. Although it is known that the remotely sensed LAI is limited by the saturation effect at high LAI values, the results of our study show that the PILOTE model simulations (DA$_{sat}$) are not considerably affected by it. In fact, the model simulations are constrained by $LAI_{sat}$ not only at high $LAI_{sat}$ values but also at low LAI values. Proper sequential

updating of $LAI_{sat}$ by PF from the early growing season reduces the negative impacts of the saturation effect that occurs in the middle of the growing season.

There were a different number and coverage of satellite images available in each of the site-years. Therefore, it is not straightforward to quantify the effect of the number of satellite images (quantity of data) and the coverage during the growing season on the yield prediction quality. Practically, regular availability of satellite data during the growing season in regions such as central Europe cannot be expected, because the sky is often cloudy. Effects of quantity and temporal distribution of satellite images could be analyzed with synthetic LAI observations [2]. Our study shows that yield prediction quality is not negatively affected when at least three data points were available for a given season. Results showed that assimilation of available data in April was more effective in KR than in SJ. The reason is linked to the different growth dynamics because winter wheat growth is delayed by about two weeks in the colder SJ region. This delay means that in April the LAI of winter wheat is still very low in SJ and not yet informative enough to improve yield prediction. It takes the mid- and late-season vegetative growth to form an LAI which is informative enough to improve predictions of winter wheat yields. DA in May and June (when stem elongation and anthesis occur) improved the yield predictions much more pronouncedly. The obvious downside of the above-mentioned delay is that potential management or policy actions as a consequence of predicted yields will also be later, too.

### 4.2. Uncertainty from the Model Inputs

Uncertainties in sowing dates and weather data can cause a large uncertainty and bias in crop yield prediction [60]. It is mirrored in the variability of LAI and yield predicted by MC (sowing date and weather data are the uncertain part of MC simulation). Prediction uncertainty associated with input uncertainty can only be reduced if there is a correlation between the inputs and the state of the assimilated variables. In the first DA step, the uncertainty did not decrease, but after 2–3 steps, the uncertainty in LAI predicted by $DA_{meas}$ decreased. Our results proved that DA using LAI data compensated for the limitations in the model inputs at coarse resolutions [61].

### 4.3. Impact of Model Errors

In applying data assimilation, we are not only faced with concerns about the accuracy of satellite data, but also about the propagation of remote sensing errors into the system, that is, how LAI simulation and the predicted yield are affected by the error in the remotely-sensed LAI. The LAI variable is assumed to correlate with the end of season yield, which is not a state variable directly used. In any plant growth model, correlation between the assimilated variable and yield has to be correctly described. This can be achieved by either model structure or parameter optimization. For data assimilation (using LAI), we need models and parameters which describe the correlation between LAI and yield. Otherwise, data assimilation will improve the assimilated LAI but not yield [32].

In general, whenever DA reduces the error in LAI prediction, grain yield prediction is expected to improve but this does not happen in all cases. For instance, in EC6_2014 MAE of LAI predicted by $DA_{meas}$ was lower than 0.2 $m^2/m^2$ and it underestimated yield by about 2200 kg/ha (difference between the median of measurements and predictions). In contrast, MAE of LAI predicted by $DA_{sat}$ was about 0.70 $m^2/m^2$ and the yield prediction error was less than 800 kg/ha. Another example is EC6_2017 where LAI predicted by MC is less accurate than that predicted by either DA procedure but the yield prediction quality is competitive. It seems that for further improving yield prediction by PILOTE+DA one needs to take into account the structure of the PILOTE model. The correlation between LAI simulation and yield can be improved by modifying the PILOTE model equations. In fact, harnessing the model by assimilating more data depends both on how much it constrains uncertainty sources (model inputs and parameter uncertainty) and on the sensitivity of yield to the uncertainty sources [33]. Factors not considered in the PILOTE model such as fertilization are likely the reason for such findings. Basically, such shortcomings of models

applied in large areas with a high degree of uncertainty is inevitable. To consider the impact of more factors on yield, more advanced (process-based) crop models can be used, but at the expense of requiring more input data.

## 5. Conclusions

We present a method for real-time prediction of winter wheat yield in low-information environments (i.e., unknown cultivars, sowing dates, crop management, uncertain weather data) in two study regions in southeast Germany. This low-information situation is common at the regional level. By using the simple crop growth model PILOTE in conjunction with data assimilation (here the particle filtering method) and uncertain model inputs, yield predictions could be substantially improved. We conclude that satellite-derived LAI observations, although noisy and in Germany often blurred by clouds, are a viable source of information. Assimilation of such data into crop models without the need to measure the LAI in the field can reduce prediction errors caused by missing management information and uncertain weather forecasts. In our case study, we found that three satellite observations before anthesis are sufficient to keep the LAI prediction on track. The application of our methodology on larger scale, where additional uncertainties from crop classification will arise, appears promising. To draw robust conclusions on the generalizability of the proposed approach also for large scales, the methodology should be tested against a higher number of site-years and satellite data. We further propose that interested researchers apply the new methodology with their crop models. We ourselves plan to implement our data assimilation algorithm into more advanced crop models than PILOTE as to investigate the impact of model structure and changing environmental conditions on yield prediction.

**Author Contributions:** H.Z.: Software, Formal analysis, Writing—original draft. T.K.D.W.: Methodology, Writing—review, and editing. J.I.: Conceptualization, Supervision, Investigation, Writing—review, and editing. W.N.: Methodology, Supervision, Writing—review, and editing. S.G.: Methodology, Writing—review, and editing. T.S.: Conceptualization, Supervision, Writing—review, and editing. All authors have read and agreed to the published version of the manuscript.

**Funding:** H.Z. was supported by the "Water-People-Agriculture" Research Training Group funded by the Anton & Petra Ehrmann-Foundation. T.K.D.W. was funded by the Collaborative Research Center 1253 CAMPOS (Project 7: Stochastic Modelling Framework), funded by the German Research Foundation (DFG, Grant Agreement SFB 1253/1 2017).

**Institutional Review Board Statement:** Not applicable.

**Informed Consent Statement:** Not applicable.

**Data Availability Statement:** The data presented in this study are available on request from the author Hossein Zare.

**Conflicts of Interest:** The authors declare no conflict of interest. The funders had no role in the design of the study; in the collection, analyses, or interpretation of data; in the writing of the manuscript, or in the decision to publish the results.

## Appendix A

### *Appendix A.1. The PILOTE Model*

The PILOTE model [34,35] is a crop model with a soil and a plant module. The soil module calculates the water balance on daily time steps using a capacity approach [53]. The core of the plant module is LAI, which in turn determines aboveground biomass and yield. LAI is assumed to be a function of temperature sum ($TT$; $°C$) and soil water stress.

$$TT_j = \sum_{i=1}^{i=j} (T_{mean\,i} - T_{base}), \tag{A1}$$

where $TT_j$ signifies temperature sum (°C) at day $j$, $T_{meani}$ is the mean temperature (°C) at day $i$ and, $T_{base}$ is the base temperature (°C), below which plant development ceases. $T_{base}$ was set to 1 °C for all cultivars.

$$LAI_j = LAI_{max} \left[ \left( \frac{TT_j - TT_e}{TT_f} \right)^{a_2} exp \left\{ \frac{a_2}{a_1} \left( 1 - \left( \frac{TT_j - TT_e}{TT_f} \right)^{a_1} \right) \right\} - \left( 1 - stress_j^\lambda \right) \right], \tag{A2}$$

In Equation (A2), $LAI_j$ and $TT_j$ stand for the green $LAI$ and temperature sum at day $j$, $TT_e$ is the temperature sum required for emergence (°C), $TT_f$ is the temperature sum required to reach maximum LAI ($LAI_{max}$) (°C), $a_1$ and $a_2$ are unitless shape parameters. The term $stress$ is a stress factor and describes the plant's sensitivity to water stress, modified by an empirical dimensionless parameter $\lambda$. For the $j$'th day, $stress$ is defined by

$$stress_j = \frac{\sum_{i=j-10}^{j} T_i^{pa}}{\sum_{i=j-10}^{j} T_i^{pm}}, \tag{A3}$$

which is the ratio between the sum of daily actual transpiration $T^{pa}$ (mm) and daily maximum transpiration $T^{pm}$ (mm) of the 10 days preceding day $j$. It varies between zero and one. If $stress$ is unity it means that there is no water stress and actual transpiration equals maximum transpiration. Although it seems not intuitive that $stress$ at unity implies well-watered conditions, we want to be consistent with the original papers [34,35] explaining the equation. To compute $T^{pa}$ and $T^{pm}$, maximum evapotranspiration ($ET_{max}$) is first calculated from crop evapotranspiration, which is determined from grass reference evapotranspiration using Hargreaves-Samani method [62]. Soil water balance and soil evaporation are then calculated at daily resolution using LAI and dynamic crop coefficient. Details on the water availability calculation and crop coefficient can be found in Khaledian et al. [35] and Allen et al. [63].

PILOTE calculates potential aboveground dry matter yield ($Y_{pot}$) from the radiation use efficiency ($RUE$; g/MJ) and the fraction of intercepted solar radiation $I$:

$$Y_{pot} = RUE \sum_{j=1}^{m} S_j I_j, \tag{A4}$$

where $S_j$ is daily solar radiation (MJ/m$^2$), and $m$ is the number of days from sowing to maturity. Daily $I$ is calculated from $LAI_j$ by:

$$I_j = 1 - e^{-kLAI_j}, \tag{A5}$$

with

$$k = \min\left( 1.0, \ 1.43 \ LAI_j^{-0.5} \right), \tag{A6}$$

where $k$ (−) is a dynamic extinction coefficient. Actual aboveground dry matter biomass yield $Y_a$ is calculated as

$$Y_a = Y_{pot} \min\left( 1.0, \frac{LAI_{av}}{LAI_{pot}} \right) \tag{A7}$$

where $LAI_{av}$ (m$^2$/m$^2$) is the mean actual LAI and $LAI_{pot}$ (m$^2$/m$^2$) the mean potential LAI (m$^2$/m$^2$) during a critical period. $LAI_{pot}$ is calculated from Equation (A2) under non-stress conditions ($stress = 1$). The critical period is the number of days between temperature thresholds, $Ts_1$ and $Ts_2$. $Ts_1$ is given by $Ts_1 = (TT_f - 100)$ °C whereas $Ts_2$ is the temperature sum (°C) necessary for maturity. $Ts_2$ is estimated during model calibration. For further details about $Ts_2$, we refer to Khaledian et al. [35].

Finally, PILOTE calculates grain yield from $Y_a$ and harvest index $HI$ (unitless). $HI$ is a function of $LAI_{av}$ given by

$$HI = \min\left[HI_{opt},\ \left(HI_{opt} - a_r(LAI_{st} - LAI_{av})\right)\right] \tag{A8}$$

here, $HI_{opt}$ is the potential harvest index, $LAI_{st}$ is the LAI threshold below which $HI$ will be affected by stress, and $a_r$ is an empirical reduction coefficient that accounts for the impact of suboptimal LAI during grain formation ($LAI_{av}$).

### Appendix A.2. Model Calibration and Evaluation Method

A region-specific calibration of the PILOTE model was performed. To this end, data from three site-years from two harvest seasons (2010, 2011) in each region were used. The remaining site-years were used for model evaluation.

The parameters $a_1$ and $a_2$ define form of the LAI curve (Equation (A2)). Calibrated values must ensure that green LAI declines to about zero at the end of the growing period (forming the bell shape). Thus, in addition to the measurement points, one dummy LAI with value zero was added at harvest time. Pre-analysis showed that it is better to optimize $a_1$ and $a_2$ separately before the final calibration (data not shown). After this step, we calibrated the model in three hierarchical steps. At first, $\lambda$, $TT_f$ and $LAI_{max}$ were estimated from observed LAI. Then, $RUE$ and $Ts_2$ were estimated from biomass data. Finally, the yield model was calibrated tuning the parameters corresponding to the harvest index ($LAI_{st}$ and $a_r$). Similar to Khaledian et al. [35], HI was limited to the higher and lower measured values. Therefore, $HI_{opt}$ was set to 0.55 and the lower limit of HI was set to 0.40, in both regions.

The parameters of the Choudhury model are $NDVI_{max}$, $NDVI_{max}$ and $\beta$. $NDVI_{max}$ and $NDVI_{min}$ should be calculated from the maximum and minimum vegetation status of the canopy during growing season. It is not realistic to take these parameters directly from satellite images when only a few images are available during the growing season. Therefore, we included them in the calibration. Before this, in each image pixels with the maximum and the minimum values were extracted. Then, the potential ranges of $NDVI_{max}$, $NDVI_{min}$ were taken to limit the possible range in the calibration. Here, we note that we did not calibrate the Choudhury model for each region. In fact, we assume that the Choudhury model parameters do not depend on the region.

We assumed that the parameters of the Choudhury model do not depend on region, and therefore did not perform a region-specific calibration to estimate them. The Choudhury model was calibrated using in situ data from 2010 to 2013. Data from 2014 to 2017 were used for model evaluation (Figure A3).

Parameters were estimated by maximizing the likelihood to observe the corresponding state variables. The distribution of the parameter uncertainty was approximated by Markov Chain Monte Carlo (MCMC) sampling [64]. To this end, a uniform distribution was assumed for each parameter, therefore, the posterior was only the proportion to likelihood. A Markov chain random walk with Metropolis acceptance probability ratio [65] was applied to decide whether the candidate parameter value could replace the previous value.

### Appendix A.3. Model Calibration and Evaluation Result

### Appendix A.3.1. PILOTE

The parameters $a_1$ and $a_2$, which determine the bell shape of the LAI curve, were fixed to values $a_1 = 4$ and $a_2 = 3$ in KR, and to $a_1 = 4$ and $a_2 = 4$ in SJ. These values ensure that LAI is zero at the end of the growing period in all site-years. The estimates of $LAI_{max}$, $TT_f$ and $\lambda$ obtained by calibration were 5.5 m$^2$/m$^2$, 1220 °C and 0.45 in KR, and 5.1 m$^2$/m$^2$, 1120 °C, and 0.3 in SJ (Table A1). LAI calibration resulted in an RMSE of 0.30 and 0.44 m$^2$/m$^2$ in KR and SJ, respectively (Figures A1 and A2), with corresponding evaluation results of 0.68 and 0.84 m$^2$/m$^2$. The parameter $RUE$ was estimated at 1.15 g/MJ

in KR which was slightly higher than SJ (1.05 g/MJ). The value for $Ts_2$ was different in each region and estimated at 1500 and 1200 °C in KR and SJ, respectively.

The estimates for $LAI_{st}$ and $a_r$, the two empirical parameters for harvest index, were different in each region. In KR, $LAI_{st}$ and $a_r$ were estimated at 3.85 m$^2$/m$^2$ and −0.15, while the estimated values in SJ were 4.80 m$^2$/m$^2$ and −0.80. Finally, the results for yield calibration showed the RMSE of 593 and 323 kg/ha in KR and SJ. RMSEs of yield for the evaluation dataset were 903 and 780 kg/ha (Figures A1 and A2).

Although a generic calibration that does not take the different cultivars into account was done for each region, the model performance with the evaluation data set was very close to calibration for both LAI and yield. PILOTE performance on LAI in the evaluation data set resulted in RMSE < 0.90 m$^2$/m$^2$ which is competitive to that in Ingwersen et al. [37] who calibrated the mechanistic process model GECROS to the same data finding RMSE = 0.74 m$^2$/m$^2$ for the LAI in the evaluation dataset. $TT_f$ is lower in SJ than KR which is related to the lower temperature in SJ. Similar to $TT_f$, RUE was lower in SJ than in KR. Research has shown that RUE depends on the environment and wheat grown in warmer regions has higher RUE [66,67]. Compared to the study by Ingwersen et al. [37], PILOTE evaluation results showed lower RMSEs (903 kg/ha and 780 kg/ha in KR and SJ, respectively), however, their GECROS model was calibrated to data from both regions simultaneously.

**Table A1.** Estimated PILOTE and Choudhury model parameters and corresponding posterior uncertainties.

| Parameter | Units | Optimized Value | |
|:---:|:---:|:---:|:---:|
| | | **Kraichgau** | **Swabian Jura** |
| $TT_f$ | °C | 1220 | 1120 |
| $LAI_{max}$ | m$^2$/m$^2$ | 5.50 | 5.10 |
| $\lambda$ | - | 0.45 | 0.30 |
| $RUE$ | g/MJ | 1.15 | 1.05 |
| $Ts_2$ | °C | 1500 | 1200 |
| $LAI_{st}$ | m$^2$/m$^2$ | 3.85 | 4.80 |
| $a_r$ | - | −0.15 | −0.80 |

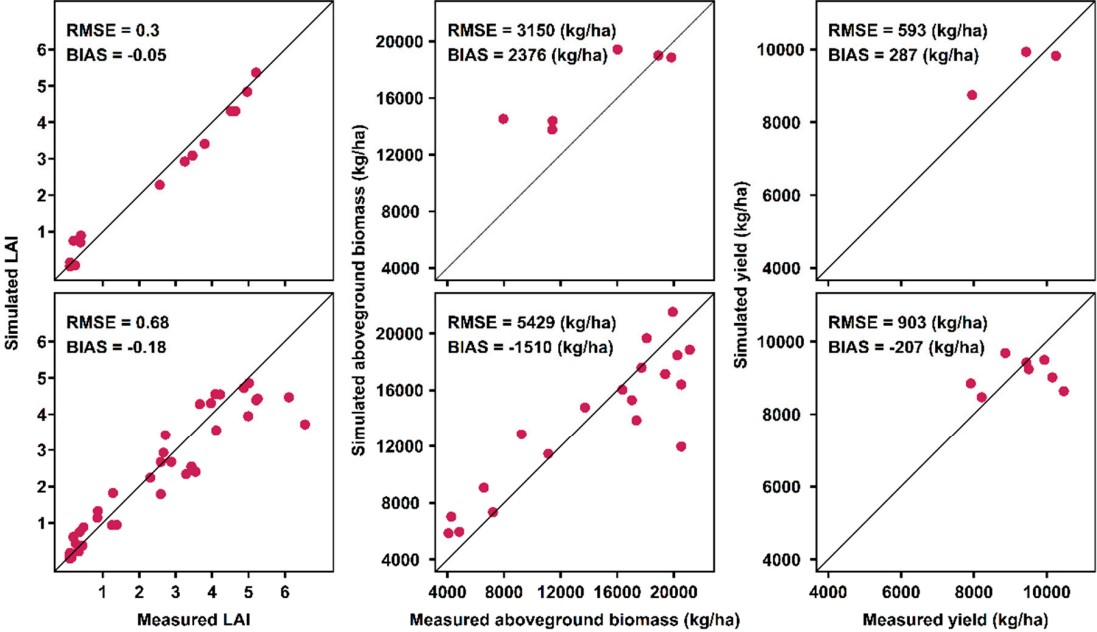

**Figure A1.** Calibration (**upper three panels**) and evaluation (**lower three panels**) results of the PILOTE model in Kraichgau (KR).

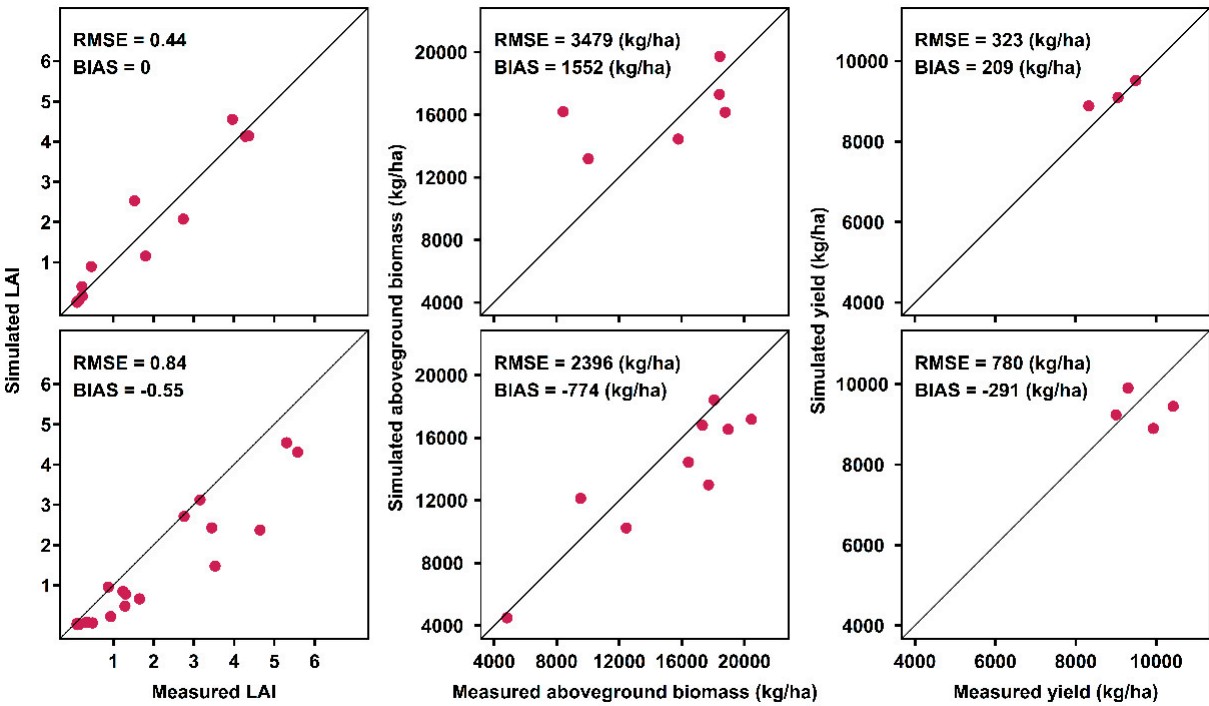

**Figure A2.** Calibration (**upper three panels**) and evaluation (**lower three panels**) results of the PILOTE model in Swabian Jura (SJ).

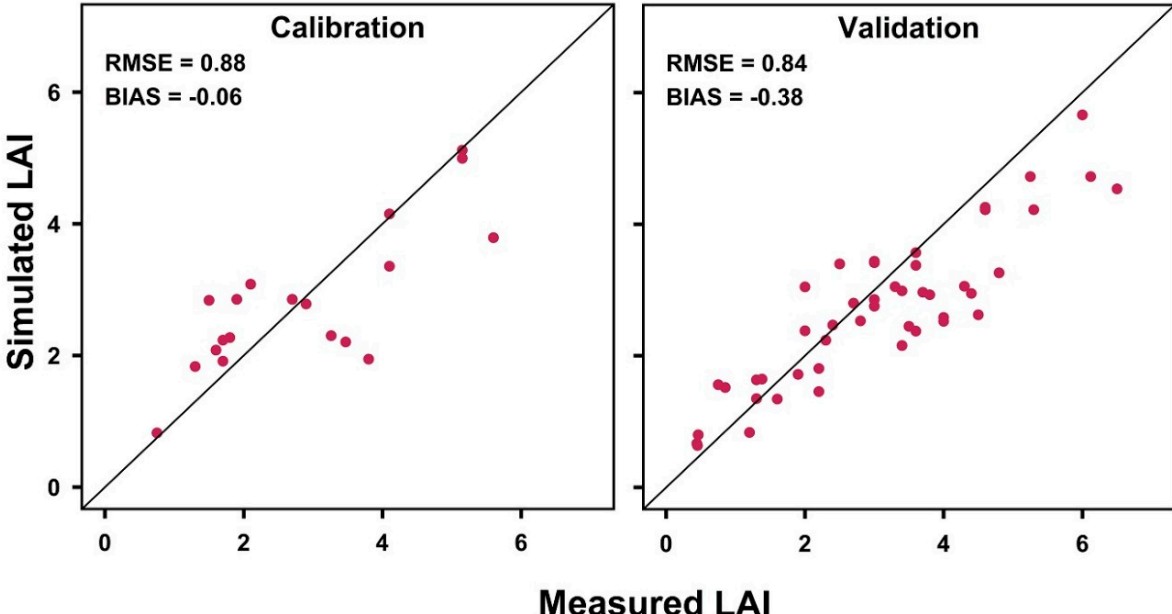

**Figure A3.** Calibration and evaluation results of the Choudhury model. Calibration is based on site-years from 2010 to 2013, evaluation on the remaining site-years.

Appendix A.3.2. Choudhury

The optimized value for the parameter $\beta$ in the Choudhury model was 0.67, and $NDVI_{max}$ and $NDVI_{min}$ were optimized at 0.91 and 0.20. The Choudhury model estimated an almost unbiased LAI and an RMSE of 0.88 m$^2$/m$^2$ (Figure A3) in the calibration data set. The accuracy of the model in the evaluation data set was close to the calibration (RMSE = 0.84 m$^2$/m$^2$). The prediction, however, was biased (bias = −0.38), mostly due to underestimation when observed LAI was higher than 4.

**Appendix B**

Figure A4 depicts assimilating the remotely-sensed derived LAI into the PILOTE model at five steps. The last LAI point is a dummy value to harness LAI to zero at the harvest time. In each step one LAI point is added to the assimilation procedure. This figure shows how adding more data in assimilation influence the RMSE and the uncertainty (variation of the cyan lines) of the LAI prediction.

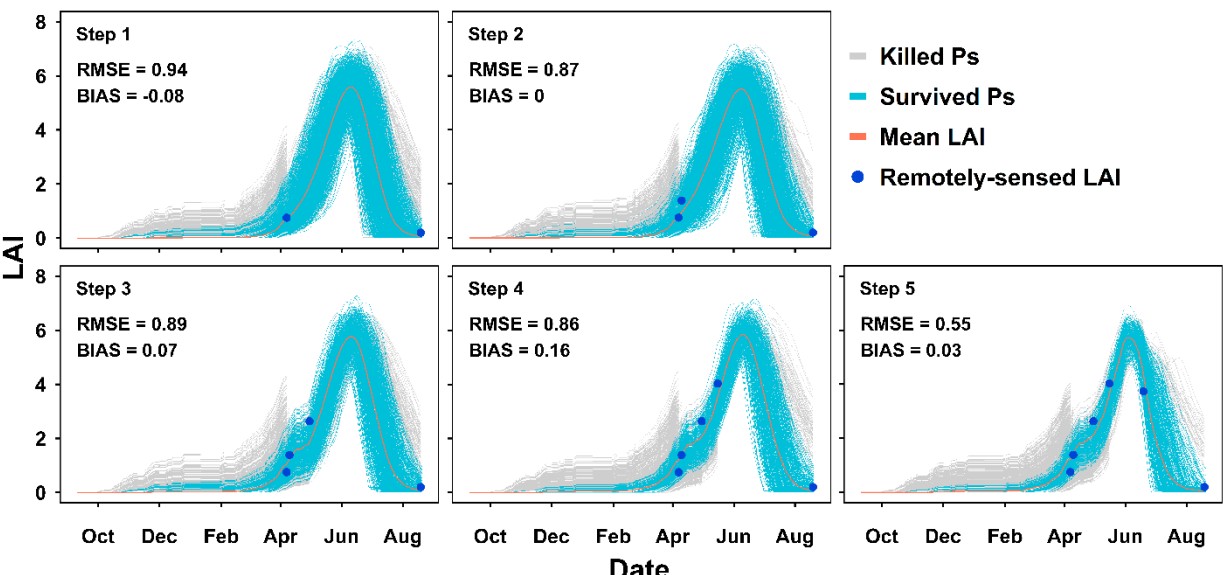

**Figure A4.** An independent run for data assimilation with 2500 number of particles using remote sensing data in five steps at EC6 in 2017. Cyan lines represent the surviving particles at each step and the gray ones shows the killed particles.

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
