# Peer review of "Combining Crop Modeling with Remote Sensing Data Using a Particle Filtering Technique to Produce Real-Time Forecasts of Winter Wheat Yields under Uncertain Boundary Conditions"

_remotesensing, doi:10.3390/rs14061360_

Round 1

Reviewer 1 Report

Here the authors have used a well studied crop yield model, PILOTE, combined with data assimilation techniques of weather and remote sensing data to provide real-time predictions of yield. The manuscript is extremely detailed, especially with all the bayesian techniques. With one exception all work here looks technically correct, and beyond that I only have some minor concerns and suggestions.

My main concern is you are missing a potentially large source of error, the parameter uncertainty. Your estimated parameters have an underlying distribution and may, or may not be, a large source of uncertainty in the final predictions. It would be good if you could incorporate this into your analysis. It’s also unlikely that the uniform distribution (L962) is appropriate for the parameter estimates. See LeBauer et al 2013, who say “The use of such vague priors often exacerbates problems with equi-finality which can produce unidentifiable parameters, as well as biologically unrealistic parameter sets that generate the right model output for the wrong reasons”. I would also recommend reading and using ideas from Dietze 2017, which explains in large detail how to incorporate all uncertainty, including parameter, into model forecasts with data assimilation techniques. 

LeBauer et al. 2013. “Facilitating Feedbacks between Field Measurements and Ecosystem Models.” Ecological Monographs 83 (2): 133–54. https://doi.org/10.1890/12-0137.1.

Dietze 2017. “Prediction in Ecology: A First-Principles Framework.” Ecological Applications 27 (7): 2048–60. https://doi.org/10.1002/eap.1589.

Minor concerns:

Section 2.5: please provide more detail on exactly what inputs went into the monte carlo simulations. Was it only the randomly generated weather data? Presumably it still used LAI values in the PILOT model aspect, were those from the in-situ observations or satellite NDVI based? (even if values were not used directly, they were still used for parameter calibration, thus affecting the MC results).

Please describe in the methods section exactly how the out of sample LAI data was used for evaluation. I only see it mentioned in the description for figure A10.

Section 2.1.2: More details are needed for the weather generation. For example: What is the need for it over using observed weather from station data? Which GCM was used as input to the MarkSim model? 

Figures 5,6: Why are there only a few discrete predictions in the growing season? Do these correspond to in-situ or Satellite LAI measurements? Since you are modeling LAI as a state variable is it not possible to have predictions regularly throughout the year? That would be a novel way to view the results and how yield  predictions change throughout the season.

The technique presented here is interesting, but is limited  in that there must be in-situ LAI observations for each field that is modeled. Please elaborate in the discussion what it would take to generalize this across large landscapes for yield estimation.

There are latex rendering errors throughout the manuscript: eg. L172-177, L236

Author Response

Dear Reviewer 1,

We greatly appreciate your deep review and comments. We took great care in addressing your comments. Please see the attached file for responses to your comments.

Kind regards,

Hossein Zare

Reviewer 2 Report

Crop yield estimation is one of the main applications of Agricultural Remote Sensing. The present paper presents a data assimilation strategy to forecast winter wheat yield through combining a crop model and remote sensing-based LAI. In this paper, there are a lot of method descriptions and details of model parameter optimization. Results show the fine contribution of LAI derived from satellite images to winter wheat yield. However, it seems that the application to remote sensing data is very limited. In the revised version, the manuscript should be improved as follows: 1. NDVI is a common vegetation index, but it has a saturation issue. When the vegetation is dense, its value is close to 1 and does not increase. So, is the LAI estimation in this paper affected by the problem mentioned above? 2. In many places throughout the paper, such as lines 172-176, 445, and 453, the authors use the incorrect "section 0" when referring to "section." This has a significant impact on reading and increases the level of incomprehensibility for readers. 3. The paper shows that RF is not restricted to Gaussian errors and can handle nonlinear changes in the system under consideration, but both the field-LAI and the satellite-derived LAI follow a Gaussian (normal) likelihood model (line 461). Are the two statements contradictory? If the two data types follow a Gaussian model, how about EnKF?

Author Response

Dear Reviewer 2,

We greatly appreciate your review and comments. We took great care in addressing them. Please see the attached file for responses to your comments.

Kind regards,

Hossein Zare

Reviewer 3 Report

Zare et al describe a data assimilation procedure to enhance the predictive performance of winter wheat yield models. This paper is well written and seems to be a useful extension of methodology.

However, I think the authors need a clearer statement of objectives and hypothesis. It isn’t clear that the focus of the paper is LAI or yield? What really matters in management/policy is yield? I understand that you only get one yield measurement per year, and multiple LAI measurements. The authors need to make clearer in the objectives what the purpose of predicting LAI is, and specifically how that relates to yield. This is described in pieces elsewhere, but would be good to have it described here.

In the methods (254-261), please provide more info on how LAI is calculated. Is only one value (mean) provided for one field per time-period? Or is it provided on a per pixel basis?

Why is sowing date treated as a random variable? It should have some, non-random, impact on yield/LAI. Is it because such information is only known in limited cases. The authors have the information to test whether sowing date actually matters in LAI/yield.

Minor Comments:

Line 17: What about weather? This would seem to be quite important as well

Line 18: Suggest deleting “the fact that the”

Line 21: “To this end”, what end is this? can the authors be more specific?

Line 22: What would sowing data indicate? When the wheat was planted (sowed)? The depth of planting? The density of planting?

Lines 34-35: Replace “either region” with “both regions”

Line 120: “However, regular accessing fine spatial resolution satellite data” isn’t clear. Suggest: However, repeat observation of fine spatial resolution data ….is often hampered by….”

Lines 136-137: True, but shouldn’t a model with robust hindcasting correlate with good real-time prediction?

Line 157: Would you not also need soil texture data (clay, silt, sand %) and organic matter data to evaluate for soil water balance?

Lines 172-177: The “0”s must be place holders. Please populate with the real section numbers.

Line 203: “the three fields” is a bit confusing, since a total of 6 fields were analyzed.

Line 211: What is “BBCH”?

Table 1: Fields with missing information were not cropped that year? Or were just not part of the analysis? In line 209, it is described that EC3-2016 is excluded. What about EC1 in 2009-2019 or EC5 in 2012-2013, for example? Why are these data missing? Other crops were grown that year? Please add this information to the caption.

Line 236: There seems to be a problem with Figure 2: “Figure 2Error! Reference source not found.”

Line 242: Suggest deleting “alternatively”

Line 243: replace “campaigns” with “satellites”

Lines 258-259: Were these the min and max values within each field? So this was calculated on a per field basis. Or is this a per pixel (i.e. temporal) min and max? Please clarify this.

Table 2: What about the earlier growing seasons? Table 1 lists fields starting in 2009-2010. Please explain the lack of images noted for these time-periods in the caption and in text.

Line 355: “All pdfs on the right-hand side of equation”, this seems to be an incomplete thought. Please address.

Line 575: Would be good to put these RMSE values in the context of mean predicted yield in kg/ha.

Line 594: “On the other side” …do the authors mean “On the other hand” or “Alternatively”?

Line 609-610: “which of course is larger”, yes this is typically true, but might not always be the case. Suggest replacing with “which is commonly larger”

Line 713: June is the last month of predictions since wheat is harvested in July? But from Table 1, some fields are harvested in August. Why not make a prediction for July then?

Figures 5 and 6: It appears that wheat yield predictions tend to increase throughout the growing season (from Apr-Jun), I assume this is due to increasing LAI over this period?

Figure 7: Please specify what the error bars represent in the caption.

Line 797: Suggest inserting “season” after “late-“, as in “late-season vegetative….”

Line 831: What about factors such as topography, and soils? Those would be inferred by the satellite imagery, but are they directly applied to the model?

Lines 868-870: This text seems to be the placeholder.

Author Response

Dear Reviewer 3,

We greatly appreciate your deep review and useful comments. We have taken great care in addressing them. Please see the attached file for responses to your comments.

Kind regards,

Hossein Zare

Round 2

Reviewer 2 Report

The modified manuscript has a great improvement in the quality. I recommend accepting this manuscript for publication in the journal, Remote Sensing.